# Capacitive Desalination and Disinfection of Water Using UiO-66 Metal–Organic Framework/Bamboo Carbon with Chitosan

**DOI:** 10.3390/nano12213901

**Published:** 2022-11-04

**Authors:** Cuihui Cao, Xiaofeng Wu, Yuming Zheng, Lizhen Zhang, Yunfa Chen

**Affiliations:** 1Center for Excellence in Urban Atmospheric Environment, Institute of Urban Environment, Chinese Academy of Sciences, Xiamen 361021, China; 2Department of Chemistry and Pharmacy, Guilin Normal College, Guilin 541119, China; 3State Key Laboratory of Multiphase Complex Systems, Institute of Process Engineering, Chinese Academy of Sciences, Beijing 100190, China; 4University of Chinese Academy of Sciences, Beijing 100049, China

**Keywords:** UiO-66, bamboo carbon, CDI

## Abstract

The zirconium-based metal–organic framework (MOF) (UiO-66)/bamboo carbon (BC) composite with chitosan was prepared using hydrothermal and impregnation methods and used for capacitive desalination (CDI) and disinfection of water. The results showed that these composites had fast ion exchange and charge transfer properties. During the CDI process, these composites’ electrodes exhibited good cycle stability, electrosorption capacity (4.25 mg/g) and excellent bactericidal effect. These carbon-based composites electrodes’ bactericidal rate for *Escherichia coli* could reach 99.99% within 20 minutes; therefore, they had good performance and were a good choice for high-performance deionization applications.

## 1. Introduction

As society progresses, environmental problems are increasingly obvious and fresh water shortages are increasingly serious [1,2,3,4]. To address this, it is necessary to recycle water resources, which requires the application of water treatment technology. At present, conventional water treatment methods include precipitation, filtration, chemistry, and advanced oxidation processes; however, these methods produce toxic secondary pollutants and carcinogens, which affect human health [5,6,7,8]. For example, chlorination and disinfection by-products are carcinogenic [9,10,11]. Therefore, new water treatment technologies need to be developed.

Capacitive deionization (CDI) technology is a new type of water treatment, which was mainly used in the early stages of desalination. As technology developed, it was gradually applied to wastewater, heavy metal removal, and bacteria removal. CDI technology is non-toxic, pollution-free, and operates simply and economically [12,13,14,15]. CDI water treatment mainly depends on the performance of electrode materials and their specific capacitance, conductivity, pore structure, etc. [16,17].

Metal–organic frameworks (MOFs), which have ordered crystal structures, high specific surface areas and good thermal stability properties [18], have been used in electrochemical sensing [19], electrocatalysis, electrochemical energy storage devices (e.g., batteries and supercapacitors) [20], gas capture and separation [21], drug delivery [22], sensing [23,24], catalysis [25], and energy storage [26].

The zirconium-based MOF material UiO-66 is composed of octahedral and tetrahedral cages and has excellent thermal, aqueous, and acid stability [27,28,29]. Based on these advantages, UiO-66 has been used as a catalytic carrier of precious metals to detect H_2_O_2_ and telomerase [30,31,32]. However, research regarding UiO-66 and its composites used in the CDI desalination process is scarce. Therefore, we hoped to synthesize the UiO-66 carbon-based material as an electrode for use in CDI desalination. Antimicrobial agents, such as chitosan, kill microorganisms on contact by physically destroying their cytoplasmic membranes, and are less sensitive to microbial resistance than traditional antibiotics [33,34,35,36,37,38]. Therefore, antimicrobial agents were loaded onto the UiO-66 carbon-based materials to possibly expand the CDI disinfection process.

In this study, carbon-based composite BC@UiO-66 was synthesized via solvothermal treatment of the UiO-66 precursor mixture and bamboo carbon (BC) [39]. Next, chitosan (CS) was loaded onto the BC@UiO-66 material using the immersion method to obtain the composites, which were defined as BC@UiO-66-CS-x. The BC@UiO-66-CS-x were prepared as CDI electrodes for the capacitive deionization disinfection of water.

## 2. Experimental

### 2.1. Materials

All chemicals, including zirconium chloride (ZrCl_4_, 98%), terephthalic acid (H_2_BDC, 99%), bamboo carbon (BC, 20 nm), methanol (CH_3_OH, 99.5%), N-Methyl pyrrolidone (NMP, 99.5%), N,N-Dimethylformamide (DMF, 99%), acetone (CH_3_COCH_3_, 99.5%), acetic acid (CH_3_COOH, 99.5%), potassium hydroxide (KOH, 90%), polyvinylidene fluoride (PVDF, 99.5%), sodium chloride (NaCl, 99.5%), and chitosan powder (99.5%) were purchased from Sigma-Aldrich and were of analytical grade.

### 2.2. Preparation of BC@UiO-66-Chitosan

UiO-66 was prepared according to the procedure reported in the literature [39,40]. First, under ultrasonic conditions, ZrCl4 (0.36 g), H_2_BDC (0.26 g), DMF (21.00 mL), and CH_3_COOH (8.60 mL) were thoroughly mixed to obtain a uniform suspension. Next, the BC suspension, which was prepared by dissolving the bamboo carbon in deionized water under ultrasonication, was added to the above uniform suspension to obtain a mixed suspension. The mixed suspension was transferred to a hydrothermal synthesis reactor and placed at a constant 120 °C for 24 h. Second, when the mixed suspension sample returned to room temperature, white crystals were obtained. After centrifugation, the supernatant was removed, and the products were repeatedly washed using acetone. After stirring, the products were dried at 65 °C for 24 h. The dried product was bamboo carbon/UiO-66 composite, named BC@UiO-66. Finally, BC@UiO-66 was modified with chitosan. To obtain chitosan solutions, 1.00, 2.00, and 3.00 g of chitosan were each dissolved in a 3 wt% acetic acid solution. Next, 0.25 g BC@UiO-66 were immersed into each chitosan solution for 24 h at 30 °C. After immersion, the mixed products were transferred to the oven to dry at 80 °C. Samples were then washed, filtered with deionized water, and transferred to the oven to dry at 105 °C. The dried products were BC@UiO-66 modified with chitosan, named BC@UiO-66-CS-x; x represented different chitosan weights.

### 2.3. Characterization

SEM images and nitrogen adsorption isotherms of the samples were measured using field emission scanning electron microscopy (FESEM, JSM-7800) (Electronics Japan-Oxford, TKY, Tokyo, Japan) and an ASAP 2020 (Micromeritics) (NSK Ltd., TKY, Tokyo, Japan), respectively. The Brunauer–Emmett–Teller (BET) method was utilized to calculate specific surface areas, pore volumes, and pore size. X-ray photoelectron spectroscopy (XPS) was performed using the Thermo Fisher Scientific ESCALAB 250Xi (Thermo Fisher Scientific-CN, Shanghai, China). Thermal decomposition of the solid was measured using SDT Q600 TA.

### 2.4. CDI Electrode Fabrication

BC@UiO-66/BC@UiO-66-CS-x, acetylene black, and PVDF (mass ratio 8:1:1) were added to the mortar. An appropriate amount of NMP was added to fully grind the mortar to obtain a slurry. Graphite paper (5 × 5 cm^2^) was coated with the slurry; samples were then transferred to dry in a vacuum drying oven at 80 °C.

### 2.5. CDI Experiments

The CDI device consisted of four main firmware: a DC power supply, a conductivity monitor, a peristaltic pump, and a CDI cell (Figure 1). The CDI cell included four main parts: end plates, soft silica gel plates, electrodes, and a spacer. The bio-contaminated water/NaCl was pumped into the CDI device at a flow rate of 12 mL/min under 1.2 V.

### 2.6. Electrochemical Measurements

The electrochemical performance of the samples was evaluated using cyclic voltammetry (CV), which was performed in a CHI 660D electrochemical workstation using a three-electrode system. The system included a saturated calomel electrode (the reference electrode), a platinum gauze electrode (the counter electrode), and a BC@UiO-66/BC@UiO-66-CS-x electrode (the working electrode). This experiment was performed in triplicate. The specific capacitances were calculated using the following formula:$$C=\int \frac{IdV}{2v\Delta Vm}$$
where *C* was the specific capacitance (F/g), *I* was the response current density (A), *V* was the voltage (V), v was the potential scan rate (V/s), and m was the mass of the electrode material (g).

### 2.7. Preparation of Microbial Cells

*Escherichia coli* (ATCC8739), broths, and agar media were obtained from American Type Culture Collection and Becton Dickinson Company (Franklin Lakes, NJ, USA). Freeze-dried bacteria were inoculated in Mueller Hinton (MH) broth and cultured at 37 °C overnight to recovery. Bacteria cells were inoculated in LB agar, cultured at 37 °C overnight, and then harvested, centrifuged, and washed with phosphate buffered saline (PBS) solution to remove residual nutrition. Cell numbers were determined using the plate colony counting method. Next, 100 µL of 10-fold dilutions was pipetted into the LB agar of a disposable sterile culture plate. Plates were cultured in a humidity incubator at a constant 37 °C overnight for colony formation.

### 2.8. In Vitro Culture

First, the BC@UiO-66/BC@UiO-66-CS-x samples, which were dispersed in sterile water under ultrasonication, were sterilized under the UV lamp for 30 min. Next, 1 mL of 106 CFU cells was pipetted into the dispersion of the sample and cultured in vitro. The in vitro culture condition was 200 rpm shaking at a constant 37 °C. After in vitro culturing for 15, 30, 60, 120, and 180 min, 0.1 mL of each suspension was pipetted into LB agar and cultured overnight at 37 °C.

### 2.9. CDI Percent Killing Calculation 

The *E. coli* suspension (10^6^ CFU mL^−1^) was prepared as the starting bio-contaminated water. During the CDI process, 0.1 mL of CDI outflow was pipetted into the LB agar of a disposable sterile culture plate and incubated overnight at 37 °C for colony formation. This experiment was performed in triplicate. The percentage kills were calculated using the following formula:$$\%\mathrm{kill}=\frac{cell\;count\;of\;control-survivor\;count\;on\;sample\;}{cell\;count\;of\;control}\times 100\%$$

## 3. Results and discussion

### 3.1. Characterization of BC@UiO-66 and BC@UiO-66-CS-x

Figure 2a is an FESEM micrograph of BC@UiO-66, which illustrates the UiO-66 crystallites’ octahedral shape and the pore structure of bamboo carbon. It also shows that UiO-66 successfully formed composite materials with bamboo carbon. In comparison, Figure 2b–d show that a thin layer of chitosan formed on BC@UiO-66’s surface, and that the surfaces of BC@UiO-66-CS-1, BC@UiO-66-CS-2, and BC@UiO-66-CS-3 became blurry; this showed that chitosan did not destroy BC@UiO-66’s structure, and indicated cooperation between BC@UiO-66 and chitosan, possibly because chitosan loaded into BC@UiO-66’s pores and onto its surfaces.

The nitrogen adsorption–desorption isotherm of BC@UiO-66 (Figure 3a) indicates that BC@UiO-66’s isotherm was a type-IV isotherm, indicating the existence of mesopores in BC@UiO-66. However, isotherms of BC@UiO-66-CS-1, BC@UiO-66-CS-2, and BC@UiO-66-CS-3 were type-I isotherms, indicating the existence of micropores in these samples. The samples’ BJH pore size distributions (Figure 3b), pore sizes, and BET surface areas are presented in Table 1. BC@UiO-66’s specific surface area and total pore volume were 453.29 m^2^/g and 0.97 cm^3^/g, respectively. However, BC@UiO-66-CS-x’s specific surface area and total pore volume were significantly lower than BC@UiO-66’s were. This was possibly associated with the chitosan loaded on the samples. Additionally, the loss of pore volume was possibly associated with the introduction of chitosan onto BC@UiO-66’s entrance and walls.

Figure 4 shows thermogravimetric behaviors of pure chitosan, BC@UiO-66, BC@UiO-66-CS-1, BC@UiO-66-CS-2, and BC@UiO-66-CS-3. Weight loss at 100 °C was mainly due to water evaporation. The large weight loss at TG 235–322 °C was mainly due to chitosan cleavage. In comparison, BC@UiO-66-CS-1, BC@UiO-66-CS-2, and BC@UiO-66-CS-3 samples exhibited similar weight loss at 300 °C. This further demonstrated that chitosan was successfully loaded on the BC@UiO-66 sample, which corresponds with the above results.

The XPS N 1s spectra of BC@UiO-66, BC@UiO-66-CS-1, BC@UiO-66-CS-2, and BC@UiO-66-CS-3 were measured using X-ray photoelectron spectroscopy (XPS). Figure 5 shows that BC@UiO-66 had no nitrogen element; however, BC@UiO-66-CS-1, BC@UiO-66-CS-2, and BC@UiO-66-CS-3 had apparent N 1s peaks, and the peak value increased with an increase in the load chitosan, corresponding to data presented in Table 2. The samples’ nitrogen contents increased from 0.8% to 2.26%; Table 2 shows that chitosan existed in BC@UiO-66-CS’s channel and on its surface.

CV experiments using BC@UiO-66, BC@UiO-66-CS-1, BC@UiO-66-CS-2, and BC@UiO-66-CS-3 electrodes samples in 0.5 M NaCl were measured at different scan rates (5, 10, 15, 25, 50, and 100 mV/s). Figure 6 illustrates a pair of wide and symmetrical redox peaks in the samples’ CV curves, indicating that the redox process was a reversible electrode process. Additionally, as the scanning rate increased from 5 mV/s to 100 mV/s, redox peaks shifted in a positive way and the peak current gradually increased, indicating a positive relationship between the scanning rate and the redox process. When Figure 6a–d were compared, it was noted that peak current gradually decreased after loading the chitosan, indicating that chitosan film on BC@UiO-66’s surface affected electron transfer, and that the greater the load, the more significant this influence would be.

Figure 7a shows conductivity variation curves with time at NaCl 200 mg/L. After four cycles, the CDI capacitance desalination process showed good stability; the CDI capacitance desalination adsorption process began quickly, slowed, and gradually tended toward dynamic balance. When positive electricity began, the ion adsorption effect of the electrode material was obvious; ions were quickly adsorbed to the corresponding electrode. Ion enrichment, which tended to be stable, increased with time. The BC@UiO-66-CS-2 electrode material’s adsorption effect was the most obvious; conductivity was reduced from 541.1 to 530.5 μS/cm, which was significantly higher than for other electrode materials. In addition, BC@UiO-66’s adsorption effect could be appropriately increased using chitosan, but weakened when the load was too large. For example, the BC@UiO-66-CS-3 electrode material’s adsorption effect was lower than that of the other electrode materials, indicating that an excessive chitosan load affected the capacitance desalination effect. Additionally, Figure 7b shows that the BC@UiO-66-CS-2 electrode material’s electrosorptive capacity was 4.25 mg/g, which was higher than that of other electrode materials, whereas the BC@UiO-66-CS-3 electrode material’s electrosorptive capacity was the lowest, which corresponds to results presented in Figure 7a.

### 3.2. Antimicrobial Activity

Figure 8 shows *E. coli* results for samples cultured in vitro for 15 min and 180 min. Figure 8a–d show that *E. coli* colony numbers significantly decreased after in vitro culture, indicating that *E. coli* was adsorbed and killed by the material during in vitro culture. Data presented in Figure 8e show that the percentage of *E. coli* killed increased as in vitro culture time increased, which further indicated that the material had a good bactericidal effect on *E. coli*. The BC@UiO-66-CS-2 material’s percentage of *E. coli* kills was the most significant, ranging from 68.94% to 98.48%, which was higher than those of the other materials were.

To further examine *E. coli*’s morphological changes during in vitro culture, it was observed using SEM after 60 min in vitro culture. *E. coli*’s morphology changed significantly from Figure 9a–e. First, its surface changed from smooth to wrinkled. Second, its length changed, and obvious defects appeared at both ends. This indicated that after 60 min of in vitro culture, *E. coli’*s surface was damaged by the physical action of the samples’ raw edges, and that chitosan’s action destroyed *E. coli*’s cell membrane, resulting in its death.

### 3.3. CDI Process

Figure 10 shows *E. coli* results when samples were cultured for 5 min and 30 min in CDI. As seen in Figure 10a–d, *E. coli* colony numbers decreased significantly after CDI, indicating that *E. coli* was adsorbed and killed by the material during the CDI process. No colonies were found after 30 min of CDI using BC@UiO-66-CS-1, BC@UiO-66-CS-2, and BC@UiO-66-CS-3 electrodes, indicating that the electrodes’ CDI disinfection effects were ideal. Data presented in Figure 10e indicates that the BC@UiO-66-CS-2 material’s kills percentage at 20 min was 100%, which was higher than those of the other materials were. This result was significantly higher than that of in vitro culture, which indicated that *E. coli* was rapid enriched to the surface of electrode materials under the action of CDI, and that *E. coli* was quickly killed under the physical effects of the electrode material and chitosan.

To further examine the adsorption and desorption of *E. coli* during CDI, after CDI, *E. coli* was observed using SEM. Figure 11a,b indicate that *E. coli* was obviously adsorbed on the surface of electrode material in the process of CDI adsorption, and *E. coli* cells appeared destroyed, with wrinkled edges. *E. coli* was obviously desorbed from the surface of the electrode material into the solution during the CDI desorption process. This indicated that the electrode materials had good adsorption, desorption and cycle stability during the CDI process.

## 4. Conclusions

In this work, BC@UiO-66 and BC@UiO-66-CS-x composites were prepared using conventional hydrothermal and impregnation methods. The composites had the advantages of large pore size, good electrical conductivity, and good electrochemical stability, which could provide faster electron transfer. Therefore, the composite materials could be used as a new type of electrode materials for CDI desalination and disinfection. The role of chitosan was not as clear; it improved desalination performance but did not affect disinfection in an evident manner. Capacitance desalination results showed that the composite materials had high electrosorption capacity and cycle stability, and capacitance disinfection results showed that the composite materials had good disinfection effects over a short time. Among the composite materials, BC@UiO-66-CS-2 had the best capacitance desalination and disinfection performances. Therefore, this carbon-based composite could promote the development of CDI and water treatment technology. In addition to bamboo carbon based UiO-66 composites, there are many other UiO-66 composites, including the solvothermal synthesis of UiO-66 nanocrystals with high surface area using acetone as the synthesis medium [41] and UiO-66/nanocellulose aerogels with hierarchical pores and low density, which were prepared using a self-crosslinking method [42]. These both have potential as new electrode materials for capacitive desalination and disinfection of water.

## Figures and Tables

**Figure 1 nanomaterials-12-03901-f001:**
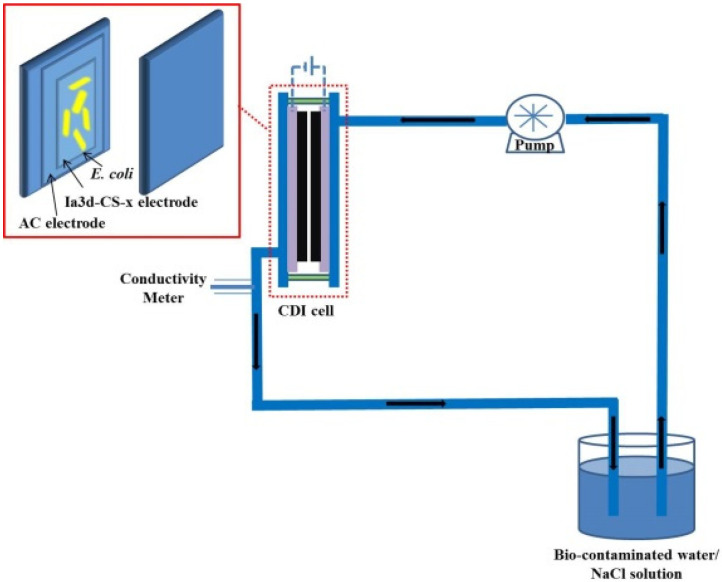
Schematic diagram of the CDI device.

**Figure 2 nanomaterials-12-03901-f002:**
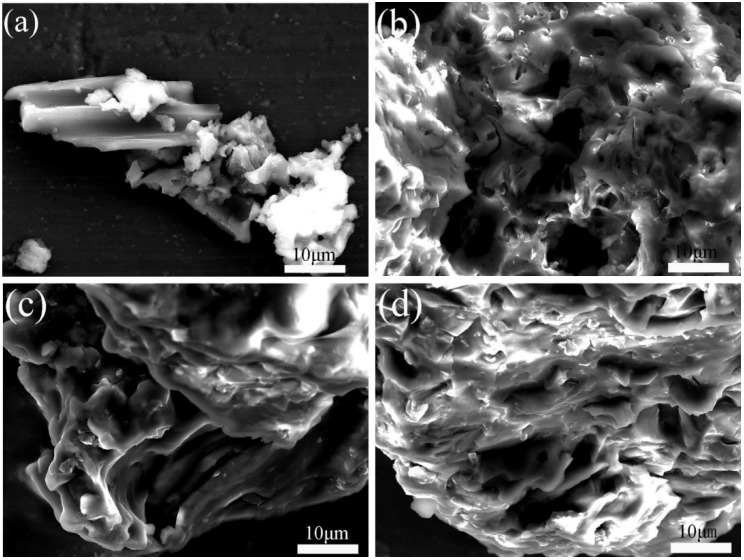
FESEM images of (**a**) BC@UiO-66, (**b**) BC@UiO-66-CS-1, (**c**) BC@UiO-66-CS-2, and (**d**) BC@UiO-66-CS-3.

**Figure 3 nanomaterials-12-03901-f003:**
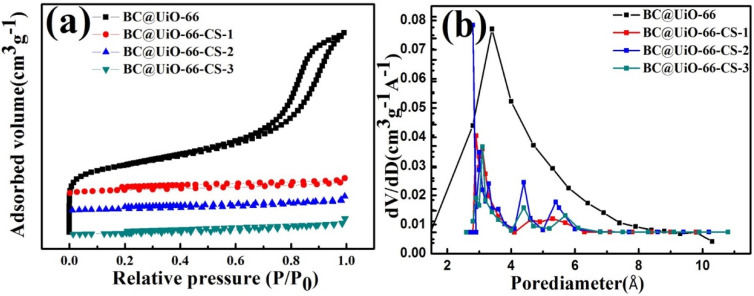
(**a**) Nitrogen adsorption–desorption isotherms of BC@UiO-66, BC@UiO-66-CS-1, BC@UiO-66-CS-2, and BC@UiO-66-CS-3 at 77 K; and (**b**) pore size distribution of BC@UiO-66, BC@UiO-66-CS-1, BC@UiO-66-CS-2, and BC@UiO-66-CS-3.

**Figure 4 nanomaterials-12-03901-f004:**
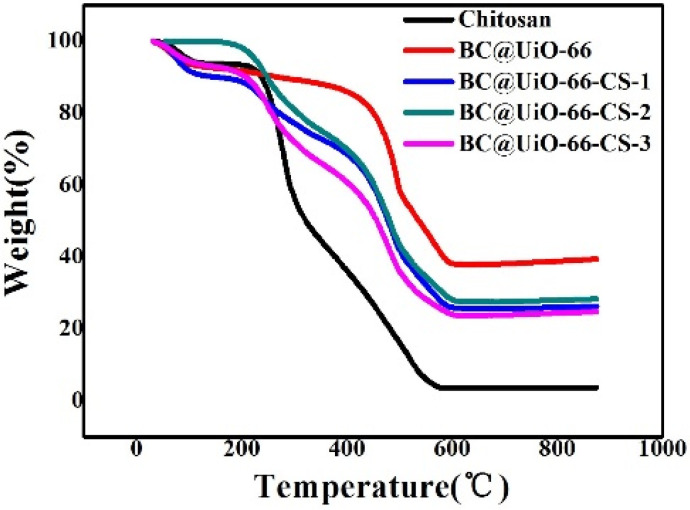
Thermogravimetric weight change curves under air atmosphere for chitosan, BC@UiO-66, BC@UiO-66-CS-1, BC@UiO-66-CS-2, and BC@UiO-66-CS-3.

**Figure 5 nanomaterials-12-03901-f005:**
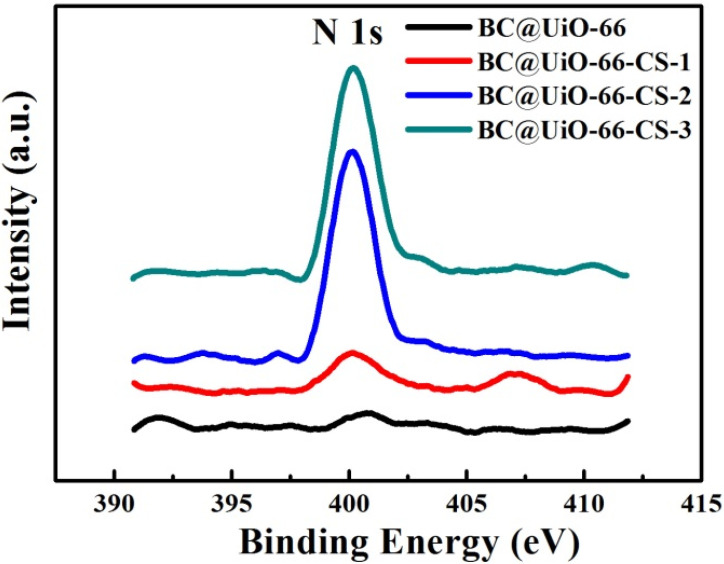
N 1s spectra of BC@UiO-66, BC@UiO-66-CS-1, BC@UiO-66-CS-2, and BC@UiO-66-CS-3.

**Figure 6 nanomaterials-12-03901-f006:**
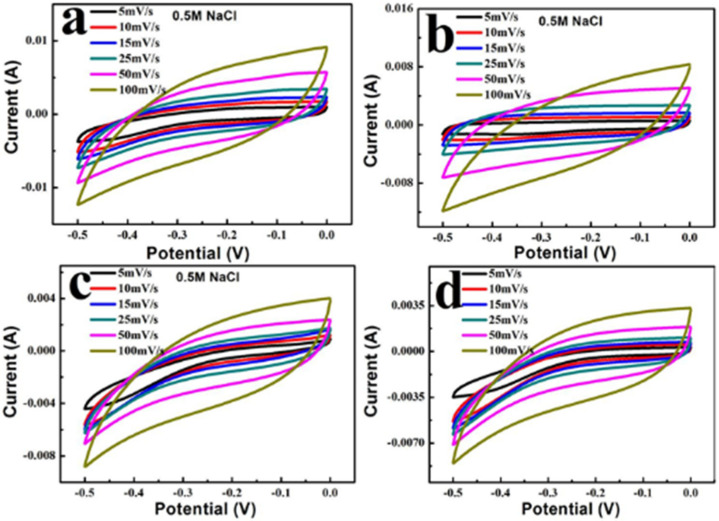
Cyclic voltammograms of (**a**) BC@UiO-66, (**b**) BC@UiO-66-CS-1, (**c**) BC@UiO-66-CS-2, and (**d**) BC@UiO-66-CS-3 at different scan rates.

**Figure 7 nanomaterials-12-03901-f007:**
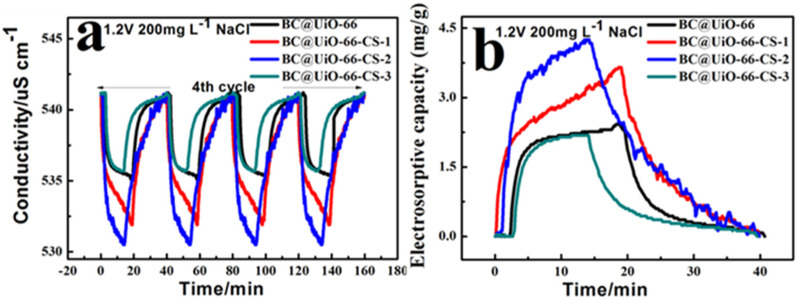
(**a**) Conductivity/time profiles of BC@UiO-66, BC@UiO-66-CS-1, BC@UiO-66-CS-2, and BC@UiO-66-CS-3 at NaCl 200 mg/L; and (**b**) electrosorptive capacity/time profiles of BC@UiO-66, BC@UiO-66-CS-1, BC@UiO-66-CS-2, and BC@UiO-66-CS-3 at 1.2 V NaCl 200 mg/L.

**Figure 8 nanomaterials-12-03901-f008:**
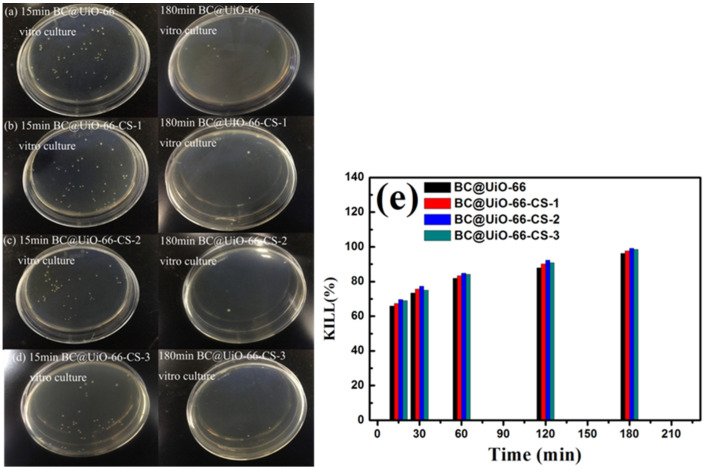
(**a**–**d**) Colony forming units; and (**e**) microbe kill rates after in vitro cultures using BC@UiO-66, BC@UiO-66-CS-1, BC@UiO-66-CS-2, and BC@UiO-66-CS-3 (100 μg mL^−1^) for 15, 30, 60, 120, and 180 min at 10^6^ CFU mL^−1^.

**Figure 9 nanomaterials-12-03901-f009:**
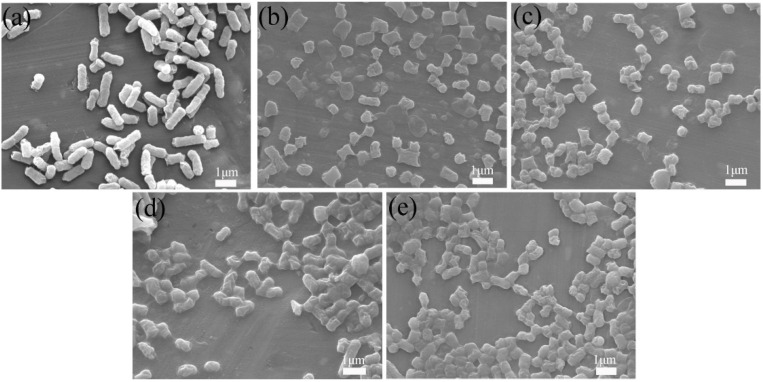
FESEM images of *E. coli*: (**a**) untreated control, (**b**) treated with BC@UiO-66, (**c**) BC@UiO-66-CS-1, (**d**) BC@UiO-66-CS-2, and (**e**) BC@UiO-66-CS-3 (100 μg mL^−1^) for 60 min.

**Figure 10 nanomaterials-12-03901-f010:**
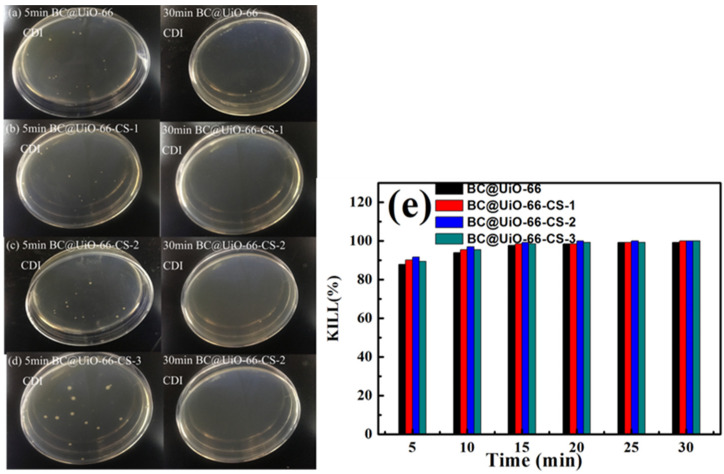
(**a**–**d**) Colony forming units; and (**e**) microbe kill rates after CDI processing using BC@UiO-66, BC@UiO-66-CS-1, BC@UiO-66-CS-2, and BC@UiO-66-CS-3 electrodes for 5, 10, 15, 20, 25, and 30 min at 10^6^ CFU mL^−1^.

**Figure 11 nanomaterials-12-03901-f011:**
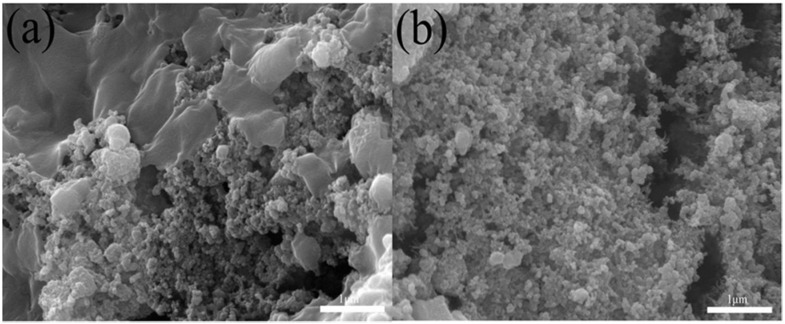
FESEM images of (**a**) the BC@UiO-66-CS electrode after the CDI process and (**b**) the regeneration process.

**Table 1 nanomaterials-12-03901-t001:** Comparison data from adsorption isotherms.

Sample	SBET (m^2^/g)	Vtot (cm^3^/g)	Dpore (Å)
BC@UiO-66	453.29	0.97	42.7
BC@UiO-66-CS-1	58.09	0.06	21.3
BC@UiO-66-CS-2	48.06	0.06	25.7
BC@UiO-66-CS-3	124.64	0.11	17.1

**Table 2 nanomaterials-12-03901-t002:** Surface elemental composition of BC@UiO-66, BC@UiO-66-CS-1, BC@UiO-66-CS-2, and BC@UiO-66-CS-3 from XPS N 1s spectra.

Sample	Atomic Concentration (%)		
	C	O	N
BC@UiO-66	92.46	7.54	0.00
BC@UiO-66-CS-1	93.16	6.04	0.80
BC@UiO-66-CS-2	85.82	12.15	2.03
BC@UiO-66-CS-3	83.77	13.97	2.26

## Data Availability

The data presented in this study are available on request from the corresponding author.

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
