# Peer review of "Capacitive Desalination and Disinfection of Water Using UiO-66 Metal–Organic Framework/Bamboo Carbon with Chitosan"

_nanomaterials, 2022, doi:10.3390/nano12213901_

Round 1

Reviewer 1 Report

Comment to the paper: “UiO-66 metal-organic framework/bamboo carbon with chitosan for capacitive desalination and disinfection of water” by Cuihui Cao et al.

General comment:

The manuscript deals with investigations on the synthesis of a material for desalination and disinfection of water.

The manuscript is suitable to be published in this journal, however some points should be addressed before publication.

Some minor language mistakes are present that should anyway be corrected.

1. Introduction

Water treatment can be performed by using several innovative approaches, such as advanced oxidation process. Please, consider the following manuscript to include in the introduction:

o   Electro-oxidation of humic acids using platinum electrodes: An experimental approach and kinetic modelling (2020) Water (Switzerland), 12 (8), art. no. 2250

o   Comparison of UV-based advanced oxidation processes for the removal of different fractions of NOM from drinking water (2023) Journal of Environmental Sciences (China), 126, pp. 387 – 395

o   Graphene shell-encapsulated copper-based nanoparticles (G@Cu-NPs) effectively activate peracetic acid for elimination of sulfamethazine in water under neutral condition (2023) Journal of Hazardous Materials, 441, art. no. 129895

2. Experimental

Please, specify if the chemicals were of analytical grade.

Please, specify the parameters investigated and their variation range.

Please, specify if investigations were carried out in duplicate/triplicate etc.

Please, specify if by-product generation was considered and monitored.

3. Results and discussion

Please, improve comparison between your findings and literature data.

Please, include by-products detected during the experiments.

Reviewer 2 Report

dear authors:

row13: Put (CDI) after desalination

row17: You don't have defined BC@UiO-66-CS-2 yet. So, i suggest to modified or to delete the sentence "among ... 20 min"

Fig3a and b please add ticks and label ticks to y axes

Fig3a: use different scales tu put in evidence diferences aong samples

Fig3b: please make this figure undestandble

Conclusions: the role of chitosan is not so decisive. It improves the desalination performance but not the disinfection in evident manner

Round 2

Reviewer 1 Report

The authors revised the manuscript according to the comments/changes suggested. The paper is suitable to be published in this journal in the current form.

Author Response

Sincerely thank you for taking time out of your busy schedule to review!

Reviewer 2 Report

Dear authors,

you added  ticks and label ticks to fig 3 (a,b) as requested but the Figures are the same ones. It is impossible to understand isotherms in fig 3a and the  curves of Fig 3b are absolutely unreadable. So please those figures must be modified heavily

Author Response

Since the value of BC@UiO-66 in Fig 3(a) was relatively large, the difference in the pictures was not obvious. As required, the curve in Fig 3 (b) had been modified.

Round 3

Reviewer 2 Report

thanks to authors fo their collaboration